



# Design of an ozone and nitrogen dioxide sensor unit and its long-term operation within a sensor network in the city of Zurich

Mueller Michael[1], Meyer Jonas[2], and Hueglin Christoph[1]

[1]Empa, Swiss Federal Institute for Materials Science and Technology, Duebendorf, Switzerland.
[2]Decentlab GmbH, Duebendorf, Switzerland.

*Correspondence to:* M. Mueller (michael.mueller@empa.ch)

**Abstract.** This study focuses on the investigation and quantification of low-cost sensor performance in application fields such as the extension of traditional air quality monitoring networks or the replacement of diffusion tubes. For this, sensor units consisting of two boxes featuring $NO_2$ and $O_3$ low-cost sensors and wireless data transfer were engineered. The sensor units were initially operated at air quality monitoring sites for three months for performance analysis and initial calibration.

Afterwards, they were relocated and operated within a sensor network consisting of six locations for more than one year. Our analyses show that the employed $O_3$ and $NO_2$ sensors can be accurate to 2-5 and 5-7 ppb, respectively, during the first three months of operation. This accuracy, however, could not be maintained during their operation within the sensor network related to changes in sensor behaviour. Hence, the low-cost sensors in our configuration do not reach the accuracy level of $NO_2$ diffusion tubes. Tests in the laboratory revealed that changes in relative humidity can impact the signal of the employed

$NO_2$ sensors similarly as changes in ambient $NO_2$ concentration. All the employed low-cost sensors need to be individually calibrated. Best performance of $NO_2$ sensors is achieved when the calibration models include also time dependent parameters accounting for changes in sensor response over time. Accordingly, an effective procedure for continuous data control and correction is essential for obtaining meaningful data. It is demonstrated that linking the measurements from low-cost sensors to the high quality measurements from routine air quality monitoring stations is an effective procedure for both tasks provided

that time periods can be identified when pollutant concentrations can be accurately predicted at sensor locations.

## 1 Introduction

Numerous gas sensors sensitive to ambient concentrations, capable of being integrated in sensor nodes or in measuring devices and available at a low price (<500 $) have entered the market in recent times. These sensors, in the following referred to as 'low-cost sensors', hold the potential for multifaceted applications and therefore attracts diverse user groups (Snyder et al.,

20 2013). Measuring devices for air quality monitoring with integrated low-cost sensors are commercially available or form part of scientific activities (e.g. Mead et al., 2013; Li et al., 2012; Piedrahita et al., 2014; Jiao et al., 2016). However, the number of studies demonstrating that low-cost sensors can provide meaningful information on ambient air quality is currently small. This fact points to sensor and/or operational issues that are not yet resolved.



There are numerous studies addressing selected topics related to low-cost sensors: Sensors or sensor principles that are utilized in this study are described e.g. by Williams et al. (2013) (metall-oxid $O_3$ sensor) and Stetter and Li (2008) (amperometric gas sensors). Mead et al. (2013) investigated the behaviour of electrochemical sensors (NO, $NO_2$ and CO). They showed that they are sensitive to the pbb level, integrated them into a sensor unit and deployed the sensor units within a sensor network in Cambridge, UK, over a three months time period. Lin et al. (2015) analysed Aeroqual series 500 $O_3$ and $NO_2$ sensors over two months concluding that they have good potential to be useful ambient air monitoring instruments in urban environments provided that they were operated in parallel. Spinelle et al. (2015) investigated the performance of different sensor models determined by field-calibrations for a cluster of $O_3$ and $NO_2$ sensors using data of five months from a semi-rural site. Focusing on sensor operation Miskell et al. (2016) presented an approach to detect drifting sensors within a sensor network based on statistical analysis and data from a small number of reference stations.

Before using air quality sensors, it should be ensured that the sensors are suitable for the intended application. This is, however, not trivial as the technical information provided by manufacturers are often not sufficient and the experience about sensor performance and behavior under real-world conditions is still limited.

An obvious application of low-cost sensor devices are sensor networks that extend or even replace traditional air pollution monitoring facilities (e.g. diffusion tubes). The lower costs of sensors compared to traditionally employed instruments may result in a higher spatial resolution of sensor networks. Requirements for this application include that the sensor network provides additional information compared to existing facilities and that the costs of operation and the information gain are balanced.

In this paper we present a case study of this application conducted in the city of Zurich. We engineered a $NO_2/O_3$ sensor node, performed sensor calibration and deployed six sensor units to form a small sensor network in Zurich. Our interest particularly focuses on the achievable performance of the sensor unit and its variation with time and procedures for monitoring the performance of sensors within a sensor network. Moreover, the study aims for the identification of factors that limit the sensor performance and areas of improvement.

## 2 Established and existing infrastructure for sensor performance tests in Zurich

### 2.1 Network of sensor units

We established a small sensor network consisting of six locations in the city of Zurich on June 10, 2015 (Fig. 1 of the supplementary materials). The developed sensor units (see section 3) were deployed and have been operated at these locations till August 2, 2016. Criteria for location selection were coverage of different pollution situations, correspondence to a routine air quality monitoring site in terms of the pollution situation (table 1) and the access to a power supply. The sensors were mounted at utility poles, at a metal fence (both at ∼3 m height) or at the railing on the flat roof of a building.



## 2.2 Routine air quality monitoring sites in Zurich

### 2.2.1 Site description

The municipal (Office for environment and health protection (UGZ), City of Zurich) and federal (Swiss Federal Office for the Environment (FOEN)) authorities operate six air quality monitoring (AQM) sites for regulatory purposes in the city of

5 Zurich (Fig. 1 of the supplementary materials). These sites are representative for pollutant situations encountered in the urban background (two sites; whereas site HEU is located in a residential area at a hillside about 200 m above the city), at urban roadsides (three sites) and at a motorway (one site) (see table 1 for details). The monitoring program comprises at all sites $NO_2$ and $O_3$ among other species. Temperature and relative humidity is measured at all sites except at HEU and SWD. We have access to the 1 minute data set for the period January 2015 to August 2016. The data set is complete except for minor data

10 gaps related to calibration periods, instrument maintenance and equipment failures (>96.4% of data at each particular station in the year 2015). Note that site SWD (located at a motorway) was replaced by an AQM site located in the urban background in January 2016.

### 2.2.2 Measured $NO_2$ and $O_3$ concentrations

Pollutant levels in Zurich are moderate (table 1) as economy is dominated by services and there is no heavy industry. Motorized

15 traffic is the main emission source for $NO_x$ accounting for 47% of the emissions in Zurich (Brunner and Scheller, 2014).

We analysed the 2015 data set in order to find temporal and location-based patterns where sensor measurements from a location of the sensor network can be linked to measurements from AQM sites. Such a link may facilitate the detection of defective sensors or even the improvement of their performance by updating their calibration parameters. The locations of the sensor network are in some distance from AQM sites. Hence, the pollutant concentrations measured by the sensors cannot be

20 assumed to be equal to that of an AQM site throughout.

Computing the differences in $NO_2$ and $O_3$ concentrations w.r.t. the urban background site ZUE reveals that time periods exist when the concentration differences are usually small in the city of Zurich (see Figures 3 and 4 of the supplementary materials). This is related to temporal patterns in source activity (e.g. traffic) and meteorology. Measurements from sites in the city (roadside and background) show that $NO_2$ and $O_3$ concentrations are most homogeneous early in the morning (01:00-

25 03:00). In contrast, the difference of $NO_2$ and $O_3$ concentrations between HEU and ZUE exhibit a different diurnal pattern. Site ZUE is a background site in the city centre whereas site HEU is located in a suburban area elevated by 200 m. Here, $NO_2$ and $O_3$ concentrations are most similar in the afternoon (11:00-17:00) when the boundary layer is high and well mixed. Average $NO_2$ concentration encountered at site HEU is slightly lower than at site ZUE. The sum of $NO_2$ and $O_3$ concentration is most homogeneous in Zurich providing an additional constraint for multi-sensor units. Consequently, $NO_2$ and $O_3$ readings

30 of a sensor located anywhere in Zurich can be expected to be similar to those of the background site ZUE on average in specific time periods depending on site characteristics.



The use of a background site such as ZUE as reference for the monitoring of low-cost sensor performance is more suitable than using a traffic site. The background site is usually less impacted by local emissions and concentrations are less variable in time.

The AQM sites in Zurich provide comprehensive information about the instantaneous $O_3$ and $NO_2$ concentration field. $O_3$ and $NO_2$ estimates for any location in Zurich derived from the measurements of the most similar AQM site in terms of traffic impact and elevation are probably better than 10 ppb on average (compare Figs. 3 and 4 of the supplementary materials). That implies that the accuracy of an additional sensor must be better than this value in order to substantially extend the information about $NO_2$ and $O_3$ concentrations in Zurich.

### 2.3 $NO_2$ diffusion tube measurements

The UGZ conducts $NO_2$ diffusion tube measurement campaigns for air quality monitoring in Zurich in varying deployments every year. The diffusion tubes (Palmes et al., 1976) are exposed for 14 days periods, respectively, and then analysed in the laboratory. The resulting concentrations are usually aggregated to an annual mean value including a correction term. The UGZ performs diffusion tube measurements at the locations of the AQM sites permanently and at the locations of the sensor network (see section 2.1) in the period 21/07/2015 to 02/08/2016. Diffusion tube measurements provide an accurate (∼2 ppb for uncorrected 14 days average concentrations) and independent reference for the $NO_2$ sensors in terms of biases of the two-weekly mean.

### 3 Developed sensor unit

### 3.1 Design

The sensor units (in short: SU) consist of two boxes (Fig. 1). Two ozone sensors (Aeroqual SM50 OZU) and a GSM module for data transmission are placed in the first box. Three $NO_2$ sensors (Alphasense NO2-B42F) and a temperature and humidity sensor (Sensirion SHT21) are placed in the second box. The NO2-B42F sensors have an ozone detaining membrane (> 500 ppm.hrs @ 2 ppm (Alphasense Ltd.); ∼1.5–2.5 years of operation for 23–38 ppb mean $O_3$ concentration). Only the second box is equipped with a ventilator as the $O_3$ sensors have an active fan. The boxes intercommunicate over a radio link. The redundant design of the boxes with respect to the low-cost sensors yields a constellation where measurements of two $O_3$ and three $NO_2$ sensors, respectively, can directly be compared with each other. Being focused on sensor testing, this is valuable information especially when the sensor units operate in distance to reference AQM sites.

Indeed, one single sensor unit (SU006) is equipped with Alphasense NO2-B4 sensors (predecessor of NO2-B42F). In contrast to the NO2-B42F sensors these are cross-sensitive to ozone.

The measuring cycle of the $O_3$ sensors lasts approximately one minute. The sensors are prompted every minute and report the latest available measurement. The $NO_2$ sensors are activated sequentially for 3 seconds periods, respectively. Measurements





are taken every 20 seconds and averaged to minute values. The resulting one minute measurements are transmitted to and stored in a central database.

The $O_3$ and $NO_2$ raw data exhibit on occasionally large positive or negative values not related to ambient gas concentrations. These spikes most likely arise from interferences between the transmitting GSM modem and the gas sensors. Their occurrences

depend on the actual location of the sensor unit and thereby on the distance and operation mode of the GSM base station the GSM modem is communicating with. These issues were reduced by increasing the data transmission interval from 1 to 12 minutes in the course of the deployment.

## 3.2    Data preprocessing

The $O_3$ and $NO_2$ raw data obtained by the SUs were preprocessed according to following schema (see Fig. 6 of the supplemen-

tary materials for an illustrative example):

First, all the measurements were forced to the full minute (leaving out the seconds) concurring with the omission of some few measurements in case of small variations in the measurement frequency. Second, measurements that are marked by sensor status flags were removed. Third, we filter the $O_3$ and $NO_2$ raw data in order to remove large outliers (i.e. spikes) from further data analysis. This filter is based on the computations of a local polynomial (R function LOESS (R Core Team, 2015)) as

well as the median absolute deviation (MAD) between measurements and the local polynomial within a moving window of 61 minutes. We define outliers as measurements deviating more than 10 times the MAD from the local polynomial. If the MAD is smaller than the 25% quantile of all differences ($|local\ polynomial\ -\ measurement|$) it is substituted by this value. This prevents the exclusion of measurements during time periods with almost no variation in the pollutant concentration.

In addition, we filter out data from the $NO_2$ sensors that directly follows the switch-on of the sensor ($\sim$24 h). Resulting one

minute data completeness of $NO_2$ sensors after preprocessing in the period 11 June 2015 to 2 August 2016 is >89% for site ETH and >96% for the other sites, respectively. One minute data completeness of $O_3$ sensors is slightly below (>82%).

## 3.3    Sensitivity and selectivity of the $NO_2$ sensors

We investigated the sensitivity and selectivity of two Alphasense B42F $NO_2$ sensors by performing tests in our laboratory. The sensors were integrated in a test device with similar design as the described SUs during the tests. The sensitivity of the

sensors were analysed by exposing them to different concentrations of $NO_2$ (0, 100 ppb), NO (0, 100 ppb), $O_3$ (0, 100 ppb), CO (0, 1 ppm) and $CO_2$ (0, 1000 ppm) as well as to relative humidity (20, 80 %) and its rate of change. The tests provided information on sensor behaviour in particular situations and revealed limitations of the achievable measurement accuracy in outdoor operation.

Test series were scheduled in order to determine the coefficients $\alpha_i$ of Eq. 1.

$$S_{NO_2} = \alpha_0 + \alpha_{NO_2} \cdot NO_2 + \alpha_{NO} \cdot NO + \alpha_{O_3} \cdot O_3 + \alpha_{CO} \cdot CO + \alpha_{CO_2} \cdot CO_2 + \alpha_{S_{RH}} \cdot S_{RH} + \alpha_{D_{RH}} \cdot D_{RH} \qquad (1)$$





$S_{NO_2}$ denotes the sensor signal, $S_{RH}$ denotes the relative humidity measured by the SUs, $D_{RH}$ denotes a term representing variations in the signal related to changes in relative humidity, and the other terms denote gas concentrations. Equation 1 has to be solved for $NO_2$ in order to assess the impact of a specific gas on the indicated $NO_2$ concentration.

Figure 2 depicts three test series. In the first and second series concentrations of $NO_2$ (0, 70 ppb), NO (0, 100 ppb), CO (0, 1 ppm) and $CO_2$ (0, 1000 ppm) were varied at about 50% relative humidity. In the third series relative humidity was varied between 40 and 60% (changes of 5% every 20 minutes) and $NO_2$ was varied between 0 and 70 ppb. The amplitude of the sensor response caused by variations in RH is of similar magnitude as the response caused by changes in $NO_2$. Additional test series showed that the signal part caused by a variation in RH depends on the magnitude and the rate of this variation. It exponentially decreases over time returning to zero. We approximate this signal by the term $D_{RH}$ in order to enhance the measuring accuracy of the sensor.

$$D_{RH}(t) = \sum_{\Delta t=0}^{-500} \Delta S_{RH}(t + \Delta t) \cdot \exp \frac{\Delta t}{\Delta t_0} \qquad (2)$$

$\Delta S_{RH}$ denotes the change in relative humidity, $\Delta t$ denotes the time shift in minutes and $\Delta t_0$ is a constant. We decided to determine $\Delta t_0$ during field calibration of the sensors as changes in RH might be effected differently in the field and in the laboratory ($\Delta H_2O$, $\Delta T$) and the precise physical cause of this signal is unknown to us. The term outlined in Eq. 2 mitigates the effect caused by changes in relative humidity but related signal parts can still significantly reduce measurement accuracy over minutes to hours. We kept the relative humidity as constant as possible in test series focusing on the determination of cross-sensitivities.

Our tests of two $NO_2$ B42F sensors did not reveal any cross-sensitivities to gases that would significantly deteriorate the measurement accuracy during ambient measurements (Fig. 2 and table 2).

## 4 Sensor calibration

The $O_3$ and $NO_2$ sensor types integrated in the SUs may both interfere with temperature and humidity (manufacturers' specifications). We found that measurements of a sensor can significantly be improved by the determination and application of an individual sensor model. Such a model maps the sensor output to an ambient pollutant concentration taking into account different environmental and pollution conditions. We calibrated the sensors being integrated in the SUs. Accordingly, the sensor model may also incorporate factors specific to the design of the sensor unit (i.e. air flow, temperature variations inside the boxes, electronics).

The effective operation of the sensors in ambient environments requires that the sensor model accurately describes the sensor behavior for a sufficiently long time period or can be updated while the sensor is in operation. The expression *sufficiently long* depends on the application type and cannot be defined in general.



## 4.1 Parallel and remote calibration

We tested two calibration approaches which differ only in the used input data: The first one (denoted as "PAR") utilizes measurements while the SUs run in parallel with reference instruments of air quality monitoring stations. Each SU operated more than three months at an AQM site before being moved to a location of the sensor network. The second approach (denoted

as "PAR/REM") is an extension of the first one. The calibration "PAR/REM" is based on the combination of two data sets: data obtained when the sensors were operated at one of the AQM sites (same as "PAR") and data from selected time periods when the sensors were operated in distance from the AQM stations within the sensor network.

Measurements from the background site ZUE were used for the remote calibration of the sensors of all the SUs irrespective of their location within the sensor network. For this purpose, time periods had to be identified when the $NO_2$ and $O_3$ concentrations

at ZUE and at the particular sensor locations were likely to be similar. This was realized by computing the 30 minutes average $NO_2$ and $O_3$ concentration ranges of two dedicated sets of AQM sites. The first set denoted as *city* contains the AQM sites RGS, SCH, STA and ZUE while the second set denoted as *background* contains the AQM sites HEU and ZUE (see table 1 for details). These sets provide plausible estimators for all the locations of the sensor network. We set the threshold for the maximum allowed difference in pollutant concentration within these sets to 5 ppb so that about 10 % of the data is selected

during the operation of the sensor network for calibration (see also Figure 4 of the supplementary materials). In 90 % of the selected time periods the set's mean $NO_2$ and $O_3$ concentrations, respectively, were in the range (2.7, 22.4) and (1.7, 60.1) ppb (set *background*) and (3.6, 31.9) and (0.5, 35.7) ppb (set *city*). Except for $O_3$ concentrations within the group *background*, the span in concentration is limited in the selected time periods. This means that the data selected during the operation of the sensor network provides better constraints for the sensor drift than for the sensitivity.

The measurements from the SUs as well as from the reference instruments were averaged to 5 minutes mean values before being utilized as model input in both approaches ("PAR", "PAR/REM"). Table 3 summarizes the calibration and operation periods of the SUs. The range of pollutant concentrations and of meteorological parameters included in the models is given by the location and the time period when the SUs ran in parallel to instruments of AQM sites.

## 4.2 Sensor models

A set of statistical models were evaluated that relate the sensor output under specific meteorological conditions (temperature, humidity) to the ambient gas concentration. Only explanatory variables were included in the sensor models that are directly measured by sensors integrated in the SUs. We analysed two model groups: Models of the first group have constant sensitivity (Eq. 3 and 4). Models of the second group assume that the sensitivity is subject to variations depending on changing meteorological conditions (Eq. 5 and 6).


$$O_3 = a_0 + a_1 \cdot S_{O_3} + a_2 \cdot S_T + a_3 \cdot S_T^2 + a_4 \cdot S_{RH} + \varepsilon_{O_3} \tag{3}$$

$$NO_2 = b_0 + b_1 \cdot S_{NO_2} + b_2 \cdot S_T + b_3 \cdot S_{RH} + b_4 D_{RH} + b_5 \cdot C_{O_3} + \varepsilon_{NO_2} \tag{4}$$

$$O_3 = (m_0 + m_1 \cdot S_{O_3} + m_2 \cdot S_{O_3}^2) \cdot (m_3 + m_4 \cdot S_T + m_5 \cdot S_T^2) + \varepsilon_{O_3}$$

$$= c_0 + c_1 \cdot S_{O_3} + c_2 \cdot S_{O_3}^2 + c_3 \cdot S_T + c_4 \cdot S_T^2 + c_5 \cdot S_{O_3}^2 \cdot S_T + c_6 \cdot S_{O_3} \cdot S_T + c_7 \cdot S_{O_3} \cdot S_T^2 + \varepsilon_{O_3} \tag{5}$$

$$NO_2 = (n_0 + n_1 \cdot S_{NO_2}) \cdot (n_2 + n_3 \cdot S_T) + n_4 \cdot D_{RH} + n_5 \cdot O_3 + \varepsilon_{NO_2}$$

$$= d_0 + d_1 \cdot S_{NO_2} + d_2 \cdot S_T + d_3 \cdot S_{NO_2} \cdot S_T + d_4 \cdot D_{RH} + d_5 \cdot C_{O_3} + \varepsilon_{NO_2} \tag{6}$$

$a_i, b_i, c_i, d_i, m_i$ and $n_i$ denote model coefficients. $S_{O_3}, S_{NO_2}, S_T$ and $S_{RH}$ denote SU sensor readings of $O_3$, $NO_2$, temperature and relative humidity, respectively. $O_3$ and $NO_2$ denote the readings of the reference instruments, respectively. $\varepsilon_{O_3}$ and $\varepsilon_{NO_2}$ denote error terms. $C_{O_3}$ is the ozone concentration provided by applying Eqs. 3 or 5 or given by reference instruments,

respectively. It is only required for the three NO2-B4 sensors of sensor unit SU006 as they are cross-sensitive to $O_3$.

The terms $b_0$ and $d_0$ denote either a constant value, a step function or a piecewise linear function. The intervals of these functions refer to the deployment periods of the sensors at different locations or to equally long time periods. The effective length of these time periods can slightly deviate from the target value if the duration of the available data set is not a multiple of the target length. Obiously, when sensors operate within the sensor network, employing a step function or a piecewise linear

function is only meaningful if remote calibration ("PAR/REM") is applied.

The term $D_{RH}$ (Eq. 2) was computed based on minute values of $\Delta S_{RH}$ and than sampled to a five minutes interval for being utilized in the sensor calibration procedure. We excluded $NO_2$ data from the parameter estimation for 30 minutes after rapid changes in RH ($|\Delta S_{RH}|$>1.65% per 5 minutes) if the range of RH exceeds 5 % within these 30 minutes. Only $O_3$ concentrations larger than 2 ppb were used for calibration of $O_3$ sensors.

We treated the $O_3$ and $NO_2$ reference measurements being free of errors in the parameter estimation. Further, we neglected any concentration differences between the instruments arising from different sampling frequencies as well as from short spatial distances.

### 4.3  Performance of the sensor models

Each SU has been operated in parallel to instruments of an air quality monitoring site for at least three months before being

moved to a location within the sensor network (see table 3 regarding calibration details). Figures 3 (a) and 4 summarize the performance of different sensor models derived from this data set (calibration "CAL"). The analysis is intended to reveal those factors the sensors most strongly depend on. Performance is indicated by means of the root mean square error based on measurements of calibrated sensors and measurements from the AQM sites in a threefold cross-validation. Initial performance of the $O_3$ and $NO_2$ sensors is about 3-5 ppb and 5-7 ppb (5 minutes mean values, residuals exceeding 50 ppb (single values)

considered as gross outliers), respectively. The $O_3$ sensors clearly outperform the $NO_2$ sensors in this stage. The scatter in the results for one individual model can be related to unequally performing sensors and to different locations (i.e. pollutant





conditions) and data periods (i.e. data partitioning during cross-validation). This shows that test procedures are required that cover a wide range of expected operation conditions.

The performance of $NO_2$ sensor models based on data when the SUs ran in parallel to instruments from AQM sites as well as when they operate remotely from AQM sites within the sensor network (calibration "PAR/REM") is depicted in Figure 3 (b). For some of the models shown in Figure 3 the terms $b_0$ and $d_0$ are time-dependent. The better performance of models with $b_0 \neq const$ and $d_0 \neq const$ compared to models with $b_0 = const$ and $d_0 = const$ suggests that the $NO_2$ sensor behaviour slightly varies over time. Piecewise linear functions referring to 60 days intervals exhibit good performance.

The application of a term that compensates effects due to changes in relative humidity improves the performance of the $NO_2$ B42F sensor models. This corroborates the impact of relative humidity changes on the sensor signal observed in our laboratory. The parameter $\Delta t_0$ for the description of the memory effect of changes in relative humidity equals about 120 minutes for best performing models. Moreover, a small dependency of temperature on the sensitivity of the sensor is likely as models described by Eq. 6 slightly outperform models described by Eq. 4.

Similar results were obtained for the models applied to the three $NO_2$ B4 sensors in SU006. However, these models include a linear ozone correction term (Eq. 4 and 6). This term is based on measurements from the reference instrument as they are more accurate than those from the integrated $O_3$ sensors.

The analysis of the $O_3$ sensor data obtained in the time period when the SU were operated at AQM sites let the sensor signal appear to be significantly, non-linearly temperature dependent. However, the relation between the sensor output and the ambient $O_3$ concentration turned out to be not stable over time. The sensitivity of the $O_3$ sensors of all SUs have progressionally decreased since start of field deployment. This behaviour is more pronounced for SUs operating at roadside locations (Fig. 5). It clearly became evident for the $O_3$ sensors of SU007 and SU008 during the about three months lasting calibration. Thus, the $O_3$ sensors of SU007 and SU008 were excluded from the assessment of the sensor models. The reason is unclear but might be related to ambient particulate matter impairing the sensors by blocking the mesh (sensor manufacturer, pers. communication). For SUs 1-6 the sensor deterioration could have been initiated as well during the calibration phase. If so, the estimated parameters of the calibration models would be influenced by this process. Likely, the decrease in sensitivity is partly compensated by the temperature dependency.

## 5   Achieved long-term sensor performance

### 5.1   Sets of processed sensor data

We produced three data sets based on different sensor models in order to investigate in detail the long-term sensor performance achieved during the operation of the sensor network (11 Jul 2015–02 Aug 2016). Data set 1 (DS 1) utilizes as sensor model a simple linear regression (Eq. 3 and 4 with only $a_0$, $a_1$, $b_0$ and $b_1$ different from 0). Its parameters were determined when the sensors operated co-located with instruments of AQM sites (calibration "PAR"). Data sets 2 and 3 (DS 2, DS 3) are based on sensor models given by Eq. 5 and 6. The parameters of the sensor models utilized for DS 2 were determined based on data when the SU were at AQM sites (calibration: "PAR"), the parameters of the sensor models utilized in DS 3 were determined based





on data from calibration and operation periods (calibration: "PAR/REM"). The term $d_0$ in Eq. 6 used for DS 3 is a piecewise linear function of 60 days intervals while it is a constant for DS 2. The features of these data sets are summarized in table 4.

Note that the mean value derived from all the redundant sensors integrated in the SUs is taken as the final one minute "SU" measurement (exceptions: SU004 and SU005 rely only on one $O_3$ sensor, SU002 relies only on two $NO_2$ sensors; compare
Fig. 6 and Fig. 7 of the supplementary materials). The application of a sensor model to the raw measurements can lead to negative concentrations and accordingly to negative one minute "SU" mean values. Negative one minute "SU" mean values were set to zero for the computation of average values for longer time periods. We focus on 30 minutes and 14 days values (corresponding to the measuring intervals of the diffusion tubes) for the SU performance analysis.

In the following, the term "measurements" related to SUs refers throughout to processed values from data sets 1 to 3. Results
of the analyses were presented referring to the IDs of the SUs or the locations where the SUs were operated. Calibration history, time of operation and construction details are related to the SU ID while encountered meteorological conditions and pollutant levels and the link of the SU measurements to measurements from AQM sites are related to the location.

## 5.2 Agreement of sensor measurements

Properly working sensors within the same SU are expected to provide comparable signals. Therefore, the measurements of the
15 redundant $O_3$ and $NO_2$ sensors were compared by means of 7 days rolling R2, RMSE and absolute differences of the mean values. The analysis is twofold and based on 30 minutes measurements being part of the data sets 1 and 3 (basic linear model and an advanced (for $NO_2$ sensors also time-dependent) model; see table 4 for details). Measurements of data set 1 can virtually be considered as raw data. Figure 6 depicts the results for the $NO_2$ sensors. The figure reveals that, in general, the three $NO_2$ sensors integrated in a SU behave rather coherent over time by means of R2 for DS 1 and DS 3. The figure related to DS 1
shows that some of the $NO_2$ sensors exhibit a drift in the signal which results in a higher RMSE and a larger difference of the mean values. These two metrics are considerably smaller for DS 3 due to time dependent parameters in the sensor model.

R2 values of neighbouring $O_3$ sensors are high on average for DS 1 and DS 3 despite of the decreased sensitivity over time (see Fig. 7 of the supplementary materials). This means that, indeed, the decreasing sensitivity of the $O_3$ sensors is probably a consequence of the encountered environment conditions (Fig 5).

## 5.3 Comparison of two-week $NO_2$ concentrations from the SUs and from diffusion tubes

The $NO_2$ sensor measurements were compared with two-weekly $NO_2$ concentrations derived from diffusion tubes. For this comparison, two-weekly sensor values have been calculated only when more than 96% of the one minute mean values were available. In few cases, the exposure time of the diffusion tubes differ from two weeks: Four diffusion tubes where exposed for 3-4 weeks and two of them were exposed only for one week. The RMSE between uncorrected two-week $NO_2$ diffusion tube
measurements and measurements from the AQM sites amounts to 2.0 ppb (N=216). Diffusion tube measurements are thus an independent and accurate reference for the $NO_2$ mean values provided by the SUs.

Figures 7 (a) depicts the time series of two-week $NO_2$ concentrations derived from diffusion tubes for the sensor network locations and the AQM sites. It shows the differences in pollution levels at the locations of the sensor network. $NO_2$ concen-





trations in Zurich are generally higher in winter than in summer, especially for sites that are moderately impacted by traffic emissions. The differences between the SU and diffusion tube measurements are depicted in Figures 7 (b) and (c) for data sets 2 (calibration: "PAR") and 3 (calibration: "PAR/REM"), respectively. There are considerable deviations between two-week mean $NO_2$ concentrations of particular SUs derived from data set 2 and from diffusion tubes. Especially the deviation of about

10 ppb observed for site WIN (SU005) is remarkable. The origin of this offset is unknown but refers to the time period between calibration and deployment. It evidences the importance of performance monitoring strategies for low-cost sensors. Agreement between diffusion tube measurements and data set 3, which includes time-dependent parameters derived from information provided by AQM sites, is clearly better. However, there are still significant deviations (up to 10-15 ppb) in particular two-week periods (e.g. values of site BUE on 11/8/16 and 25/8/16).

Replacement of diffusion tubes by low-cost sensors would be an interesting option due to the high temporal resolution of sensors. However, further improvements of the sensors and the processing procedures are required to reach the accuracy and reliability level of diffusion tube measurements.

## 5.4    Linking SU measurements to measurements from AQM site ZUE

We compared $NO_2$ and $O_3$ measurements at locations of the sensor network (SUs) as well as the AQM site network with

measurements from AQM site ZUE by computing the concentration differences. The analysis is twofold: First, it takes into account all the data obtained during the operation of the sensor network. Second, it utilizes only the data from time periods when the concentration ranges within the respective AQM site groups (*background* or *city*) are below 5 ppb (Figures 8 (a) to (c)). We expect similar distributions of the differences for SUs and AQM sites that are comparable in terms of site type.

Figure 8 (a) depicts $NO_2$ data from data set 2 (calibration: "PAR") showing large offsets (> 5 ppb) between the measured

and the expected mean for sites PFI and WIN in the selected time periods. Note that AQM site SWD is not part of any AQM site group confirming the effectiveness of this analysis approach. As expected, the fit of data set 3 (calibration: "PAR/REM") is clearly better because the data from the selected time periods was used for the determination of sensor model parameters (Fig. 8 (b)). The scattering of the concentration differences for the SUs is significantly larger than for the AQM sites pointing to a lower accuracy of the sensors.

The potential of monitoring and improving low-cost sensors based on measurements from AQM sites is illustrated in Figure 9. The Figure shows the variation of the parameter $d_0$ which is time-dependent for $NO_2$ sensors in DS 3. It accounts for temporal variations of the zero point offset. In addition, this figure shows the comparison of the concentration differences between two-week diffusion tube measurements and two-week means of $NO_2$ sensor measurements of DS 2 (no time-dependent parameters) with the parameter $d_0$ of DS 3. The good correspondence of these two independent values demonstrates that using

data from AQM sites for the identification of time periods when pollutant concentrations differ only little at specific locations and thus can accurately be predicted is a feasible approach for linking data from AQM sites and from sensors. This is a basic strategy for the performance monitoring of sensors. Further, it shows that sensor measurements can even be improved based on this strategy. A prerequisite remains that the sensitivity of the sensor is sufficiently stable as minimum values are better constrained by linking measurements from different locations than the span.



The agreement of the $O_3$ sensor measurements of data set 3 is rather weak but concurs with the reduction of sensitivity with advancing operation time (Fig. 8 (c), see also Fig 5).

## 5.5 Re-calibration of SUs

At two locations of the sensor network the SU were replaced after six months of operation: at location PFI (SU004 by SU007) and at location GES (SU006 by SU008) (see table 3 for dates). SU004 and SU006 have run in Zurich for about one year in total at this time. Directly after their replacement they were operated at the AQM site SCH for one month more. This data was used to analyse the sensor performance after one year of operation by analysing the fit between the measurements of the calibrated sensors (DS 2: intial calibration) and the measurements from the AQM site.

The agreement of the four $O_3$ sensors with the measurements from the AQM site was poor in terms of RMSE (Fig. 10; compare also Fig. 5). However, the correlation between measurements from the $O_3$ sensors and the reference instrument was still between 0.59 and 0.91.

The RMSE of the $NO_2$ sensors amounted to 8.0-11.7 ppb and 6.1-9.2 ppb for SU004 (NO2-B42F) and SU006 (NO2-B4, applied $O_3$ correction based on measurements from the $O_3$ instrument of site SCH), respectively. Correlation was above 0.89 for all the sensors. Figure 10 show that the sensitivity of the three examined NO2-B42F and NO2-B4 sensors, respectively, remained rather stable over the one year period. The linear dependency between sensor and reference measurements can essentially be described by an offset of several ppb and a slope of ~0.8–0.9. The offsets agree very closely with the $d_0(t)$-terms included in the sensor model described by Eq. 6 on which $NO_2$ values of data set 3 are based (Fig. 9). The analysis gives clear evidence for the necessity of ongoing performance analysis strategies when using such sensors for air quality monitoring.

## 5.6 Comparison of diurnal $NO_2$ variations

The locations of the sensor network were selected in order to have a corresponding AQM site in terms of the pollution situation. This facilitates the comparison of patterns of diurnal variation in pollutant concentration at different locations. Figure 11 separately depicts the diurnal variation for working days and weekends/public holidays for 2 corresponding AQM and sensor network site pairs, respectively, for the period May to July 2016. At this time the SUs have been operated at these locations for one year. The patterns of diurnal variations in $NO_2$ are as expected given the typical diurnal cycle of the traffic activity and boundary layer stability. Obviously, the accuracy of a single 30 minutes $NO_2$ concentration provided by the sensors cannot be quantified in this configuration.

## 6 Conclusions

We find that $O_3$ and $NO_2$ low-cost gas sensors can provide concentration measurements with an accuracy of a few ppb (3-8 ppb) in the first 1-3 months of operation. Comparisons with diffusion tube measurements and measurements from AQM sites revealed that this accuracy could not be maintained during the entire one year network deployment due to changing response behaviour of the sensors. Several issues were encountered related to the employed sensors that effect a temporary (~hours)



or persistent decrease of sensor accuracy. Hence, the application of performance monitoring strategies is a prerequisite of operating low-cost sensors with such properties in order to be able to assess the quality of the data.

All the sensors of the employed types require individual calibration. Sensor calibration next to reference sites is time-consuming and requires infrastructure. Moreover, the quality of the compiled calibration data set for the model parameter

estimation depends on the prevalent ambient conditions (i.e. encountered pollutant concentrations, meteorology).

The signal of the employed $NO_2$ sensors is heavily impacted by changes in relative humidity. This effect can be reduced to a certain degree by the application of a correction function but still limits the achievable accuracy of the sensors. This issue points to the necessity of an improved mathematical description of the sensor based on its working principle in order to describe sensor behaviour in more detail. More sophisticated sensor models may facilitate calibration as its parameters can possibly be

constrained with less effort than is required when applying regression models.

$NO_2$ and $O_3$ concentration predictions can be derived on the level of a few ppb in specific time periods for many locations in Zurich using the data from the AQM site network which covers a wide range of different pollutant situations. This feature can be a substantial factor for an effective monitoring of the sensor performance in low-cost sensor networks. Moreover, such data can be used for the remote correction of sensors. In our study this procedure was shown to improve the results of $NO_2$ sensors.

Enhanced performance is required for future low-cost sensors but the extent of achievable improvements is open. Thorough sensor testing procedures adequate to the designated application as well as efficient performance monitoring strategies as demonstrated in this study will remain important features in sensor based ambient air quality monitoring.

*Acknowledgements.* We thank Jürg Brunner, Markus Scheller, Noel Rederlechner and Barbara Siegfried from the Environment and Health Protection Department of the City of Zurich for providing measurements from AQM sites, for their contribution to and performance of the

diffusion tube measurements and for their support related to sensor calibrations. Moreover, we acknowledge the support of Michael Meindl from ETH Zurich during the diffusion tube campaign. Peter Graf and Yannick Stöeferle, Empa, supported the sensor testing in our laboratory. The project was financially supported by the Swiss State Secretariat for Education, Research and Innovation (SERI) and the Swiss Federal Office for the Environment (FOEN).



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



**Table 1.** List of the air quality monitoring (AQM) sites and the sensor network sites. $\overline{NO_2}$ and $\overline{O_3}$ denote the NO$_2$ and O$_3$ annual mean concentrations in 2015, respectively. AQM sites in Zurich are operated by the Environment and Health Protection Department (UGZ, City of Zurich) and the Federal Office for the Environment (FOEN).

| Name | Network | Elevation [m a.s.l.] | Pollutant situation | Location | Group | $\overline{NO_2}$ [ppb] | $\overline{O_3}$ [ppb] |
|------|---------|-----------|---------------------|----------|-------|------|------|
| HEU | AQM (UGZ) | 610 | urban background | elevated, hillside | Background | 9.4 | 28.5 |
| RGS | AQM (UGZ) | 433 | urban roadside | city | City | 26.2 | 17.6 |
| SCH | AQM (UGZ) | 415 | urban roadside | city | City | 23.3 | 20.5 |
| STA | AQM (UGZ) | 445 | urban roadside | city | City | 17.5 | 22.4 |
| SWD[1] | AQM (UGZ) | 430 | motorway | city | | 24.4 | 18.8 |
| ZUE | AQM (FOEN) | 408 | urban background | city | Background / City | 16.2 | 24.0 |
| BUE | EMPA | 408 | urban roadside | city | City | | |
| ETH | EMPA | 535 | urban background | elevated, hillside | Background | | |
| GES | EMPA | 408 | urban roadside | city | City | | |
| PFI | EMPA | 402 | urban roadside | city | City | | |
| STB | EMPA | 409 | urban background | city | Background | | |
| WIN | EMPA | 488 | urban roadside | city | City | | |

[1] SWD was replaced by an AQM site located in the urban background in January 2016.

**Table 2.** Numeric values of the parameters $\alpha_i$ (Eq. 1) of two models derived from data of the three test series depicted in Figure 2. The parameters $\alpha_{CO}$, $\alpha_{CO_2}$ and $\alpha_{NO}$ were omitted in the first model and estimated in the second model. r denotes the pearson correlation coefficient and N the number of one minute measurements used for the parameter estimation. Units for coefficients $\alpha_i$ are [mV/ppb], for RMSE is [mV].

| Sensor (NO2_XX) | const | $\alpha_{NO_2}$ | $\alpha_{D_{RH}}$ ($\Delta t_0 = 120d$) | $\alpha_{CO}$ | $\alpha_{CO_2}$ | $\alpha_{NO}$ | $RMSE$ | $r$ | $N$ |
|------|-------|------|------|------|------|------|------|------|------|
| 00 | -15.91 | 0.1593 | 0.70 | - | - | - | 0.93 | 0.98 | 5452 |
| 00 | -15.93 | 0.1606 | 0.70 | 1.21E-04 | 1.87E-07 | -1.75E-03 | 0.94 | 0.98 | 5235 |
| 01 | -0.09 | 0.1547 | 0.63 | - | - | - | 0.92 | 0.98 | 5136 |
| 01 | -0.13 | 0.1548 | 0.63 | 1.35E-04 | 1.81E-07 | -2.04E-04 | 0.92 | 0.98 | 4932 |





**Table 3.** Operation of the sensor units (SU). PAR denotes calibration of sensors of the SU while they were co-located with instruments from AQM sites, 2$^{nd}$-PAR denotes check of the sensor performance after operation time (co-location with instruments from AQM sites), and REM denotes operation of the SUs at a location of the sensor network. $T_{amb}$, $T_{sensor}$, O$_3$ and NO$_2$ indicate the range of ambient temperature measured by instruments from the AQM site, temperature inside the SU, ozone and nitrogen dioxide, respectively, during the calibration periods. SU006 is equipped with Alphasense NO$_2$ B4 sensors, while the others are equipped with Alphasense NO$_2$ B42F sensors.

| SU ID | Location | Date | Date | Mode | $T_{amb}$ [°C] | $T_{sensor}$ [°C] | O$_3$ [ppb] | NO$_2$ [ppb] |
|---|---|---|---|---|---|---|---|---|
| 1 | RGS | 2015-02-06 | 2015-05-18 | PAR | (-5.6, 26.7) | (-3.2, 30.1) | (2.0, 60.4) | (1.7, 147.7) |
| 1 | BUE | 2015-06-10 | 2016-08-02 | REM | | | | |
| 2 | ZUE | 2015-02-02 | 2015-04-28 | PAR | (-4.7, 23.9) | (-5.0, 30.1) | (2.0, 66.4) | (2.2, 61.3) |
| 2 | ETH | 2015-04-28 | 2016-08-02 | REM | | | | |
| 3 | ZUE | 2015-02-02 | 2015-05-19 | PAR | (-4.7, 27.0) | (-5.1, 34.3) | (2.0, 84.6) | (1.6, 61.3) |
| 3 | STB | 2015-06-10 | 2016-08-02 | REM | | | | |
| 4 | RGS | 2015-02-06 | 2015-05-06 | PAR | (-5.6, 24.7) | (-3.2, 32.6) | (2.0, 60.4) | (1.7, 143.8) |
| 4 | PFI | 2015-06-10 | 2016-02-05 | REM | | | | |
| 4 | SCH | 2016-02-05 | 2016-03-14 | 2$^{nd}$-PAR | | | | |
| 5 | SCH | 2015-02-06 | 2015-05-18 | PAR | (-5.1, 29.4) | (-4.7, 34.1) | (2.0, 77.8) | (0.9, 85.2) |
| 5 | WIN | 2015-06-10 | 2016-08-02 | REM | | | | |
| 6 | SCH | 2015-02-06 | 2015-05-18 | PAR | (-5.1, 27.4) | (-4.7, 33.0) | (2.0, 77.8) | (0.9, 85.2) |
| 6 | GES | 2015-06-10 | 2016-02-05 | REM | | | | |
| 6 | SCH | 2016-02-05 | 2016-03-14 | 2$^{nd}$-PAR | | | | |
| 7 | SCH | 2015-10-27 | 2016-02-05 | PAR | (-5.9, 20.3) | (-5.7, 21.9) | (2.0, 39.4) | (0.3, 75.9) |
| 7 | PFI | 2016-02-05 | 2016-08-02 | REM | | | | |
| 8 | SCH | 2015-10-27 | 2016-02-05 | PAR | (-5.9, 20.3) | (-5.5, 22.8) | (2.0, 39.4) | (0.3, 75.9) |
| 8 | GES | 2016-02-05 | 2016-08-02 | REM | | | | |



**Table 4.** Description of the computed data sets. $f_{O_3} \sim t$ and $f_{NO_2} \sim t$ denote wether the model is time-dependent. The descriptors $PAR$ and $REM$ denote whether the parameters of the sensor model are determined based on data from co-location periods or also from data during sensor network operation.

| Data set | $O_3$ model | $f_{O_3} \sim t$ | $NO_2$ model | $f_{NO_2} \sim t$ | Calibration data |
|---|---|---|---|---|---|
| 1 | Eq. 3: $a_2 = a_3 = a_4 = 0$ | no | Eq. 4: $b_2 = b_3 = b_4 = 0$ | no | PAR |
| 2 | Eq. 5: $c_0 = const, c_2 = c_5 = 0$ | no | Eq. 6: $d_0 = const, \Delta t_0 = 120\ min$ | no | PAR |
| 3 | Eq. 5: $c_0 = const, c_2 = c_5 = 0$ | no | Eq. 6: $d_0$: piece-wise linear function (interval length of 60 days), $\Delta t_0 = 120\ min$ | yes | PAR/REM |

**Table 5.** Comparison of the $NO_2$ concentrations provided by the SU with the diffusion tube concentrations for each site of the sensor network. The row denoted by AQM summarizes the comparison between measurements from instruments of AQM sites and diffusion tube measurements. Data set refers to the sensor data processing strategy outlined in table 4. $RMSE$, $r$ and $N$ denote the root mean square error, the pearson correlation coefficient and the number of samples, respectively.

| Data set | site | RMSE [ppb] | r | N |
|---|---|---|---|---|
| 2 | BUE | 5.4 | 0.37 | 27 |
| 3 | BUE | 6.9 | 0.26 | 27 |
| 2 | ETH | 3.2 | 0.86 | 15 |
| 3 | ETH | 5.0 | 0.93 | 15 |
| 2 | GES | 6.6 | -0.20 | 9 |
| 3 | GES | 2.0 | 0.65 | 9 |
| 2 | PFI | 5.6 | 0.80 | 26 |
| 3 | PFI | 3.0 | 0.78 | 26 |
| 2 | STB | 6.9 | 0.74 | 27 |
| 3 | STB | 2.0 | 0.90 | 27 |
| 2 | WIN | 14.6 | 0.68 | 26 |
| 3 | WIN | 2.1 | 0.92 | 26 |
| - | AQM | 2.0 | 0.97 | 216 |




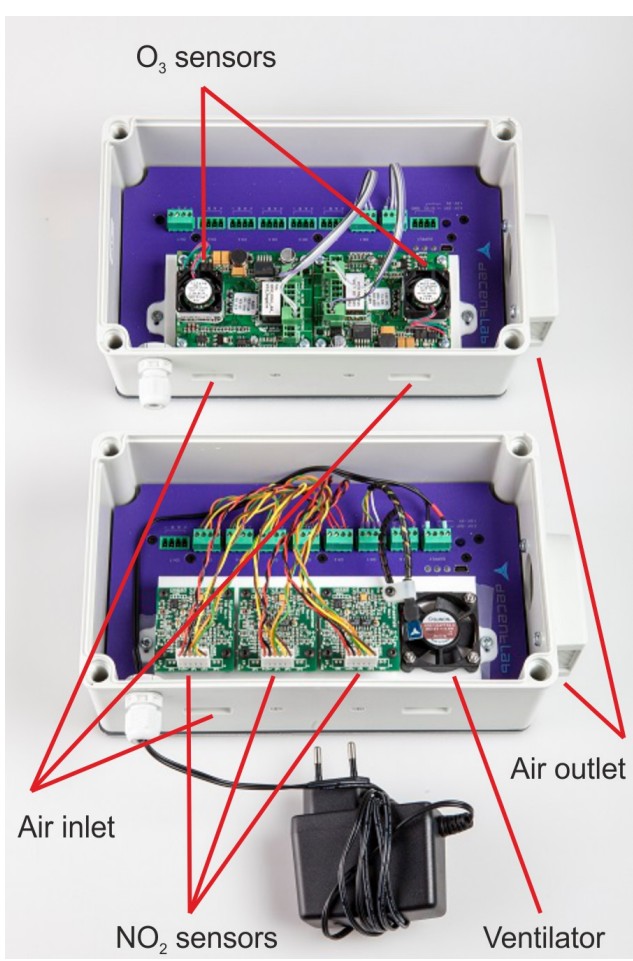

**Figure 1.** Interior view of the engineered sensor unit (top: box 1, bottom: box 2).





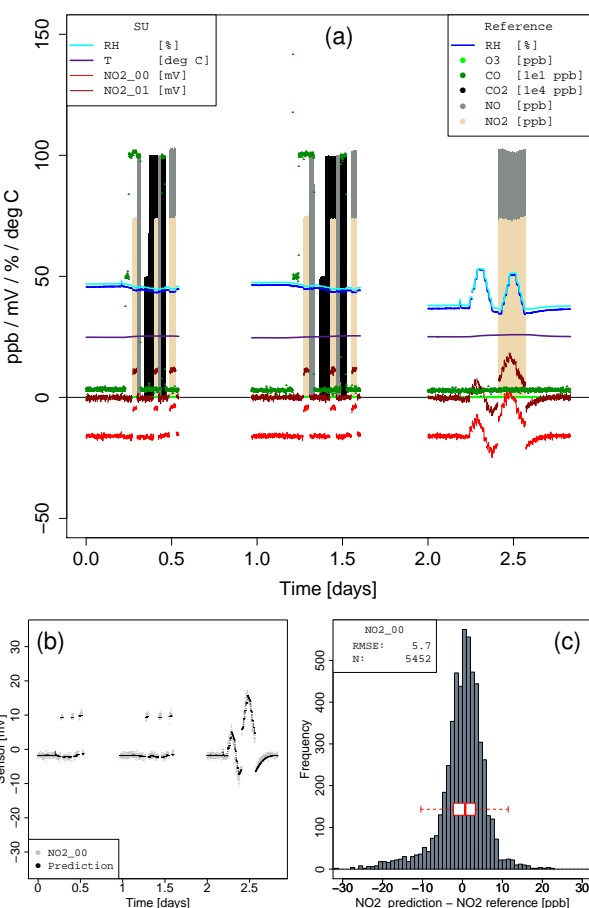

**Figure 2.** (a) Time series of sensor signals (NO2_00 / NO2_01) [mV], gas concentrations [ppb], temperature [$^\circ$C] and relative humidity [%]. Time is indicated by days. The gaps between the three parts may last longer than depicted. CO concentration is scaled by a factor 10, $CO_2$ concentration is scaled by a factor 10'000 to enhance the clarity of the figure. Note that $O_3$ concentration is constant at 0 ppb in this series. (b) Measured and predicted sensor signal (Eq. 1 with $\alpha_{NO_2} \neq 0$, $\alpha_{D_{RH}} \neq 0$ and $\alpha_i = 0$; mean sensor value subtracted). (c) Histogram of differences between $NO_2$ predictions (Eq. 1 as specified in (b), solved for $NO_2$) and measurements of the reference instrument.





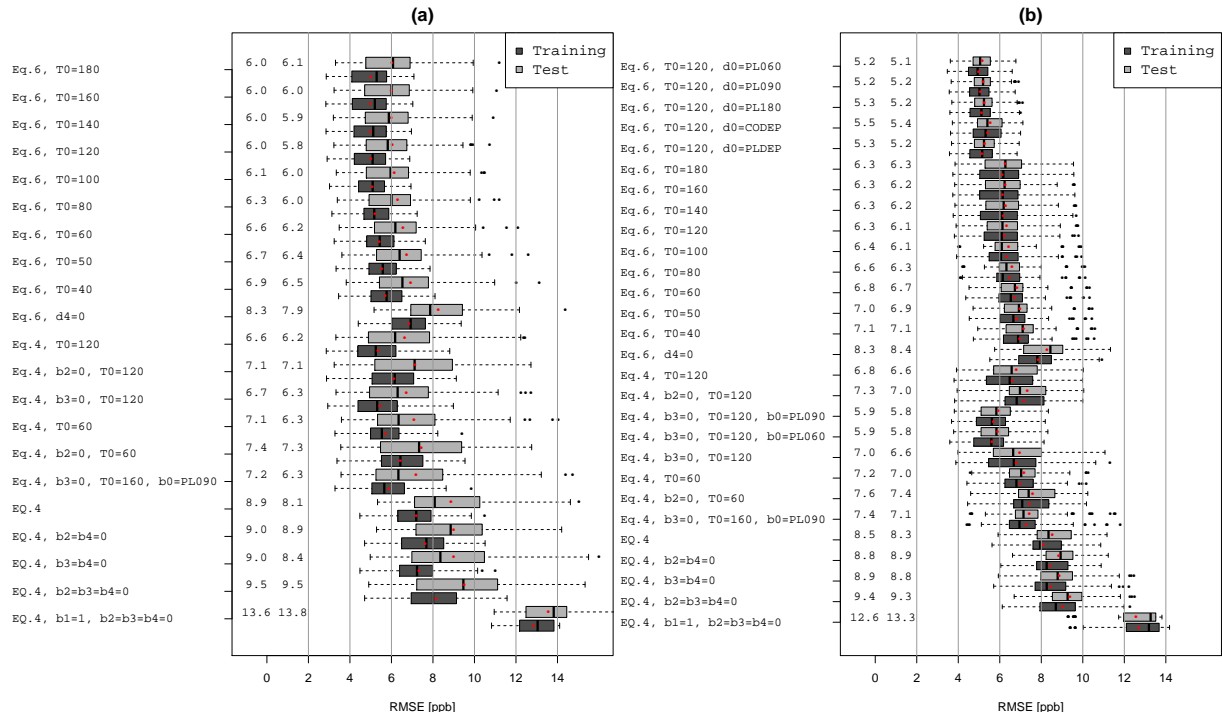

**Figure 3.** Performance of $NO_2$ sensor models based on Eqs. 4 and 6. (a) Model parameters estimated based on data when the SUs ran in parallel with instruments of AQM sites ("PAR"). The threefold cross-validation utilizes the thirst, second and third part of the data as test data set while the remaining data is part of the training data set, respectively (threefold cross-validation, three permutations). (b) Model parameters estimated based on data when the SUs ran in parallel with instruments of AQM sites and remote ("PAR/REM"). Data of every third day is part of the training data set, data of the remaining days are part of the test data set (threefold cross-validation, three permutations). Only $NO_2$ B42F sensors were included in (a) and (b). The terms $PLX$, $PLDEP$ and $CODEP$ refer to the parameters $b_0$ and $d_0$ in Eq. 4 and 6. $PLX$ denotes a piecewise linear function with an interval length of about X days, $PLDEP$ denotes a piecewise linear function with the intervals equalling the deployment periods and $CODEP$ denotes a step-wise function with the intervals equalling the deployment periods. $T0$ refers to the parameter $\Delta t_0$ in Eq. 2. The two numbers indicated for each model refer to the mean (red dot) and the median RMSE.





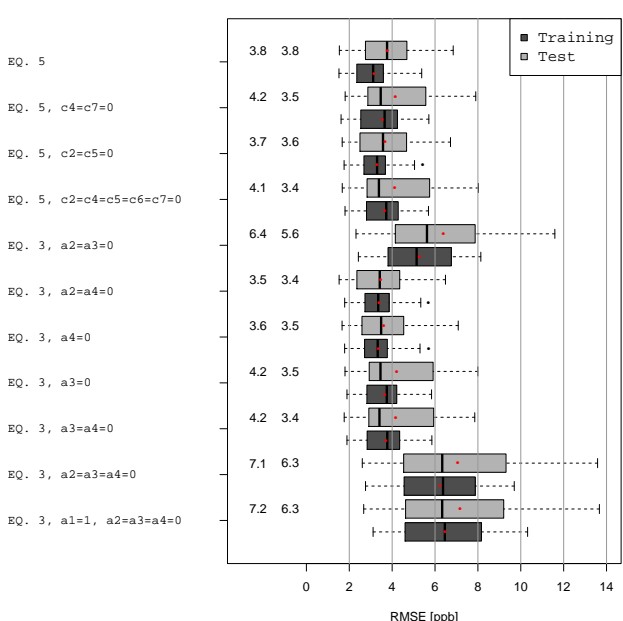

**Figure 4.** Results for the $O_3$ sensors analogue to Fig. 3 (a).





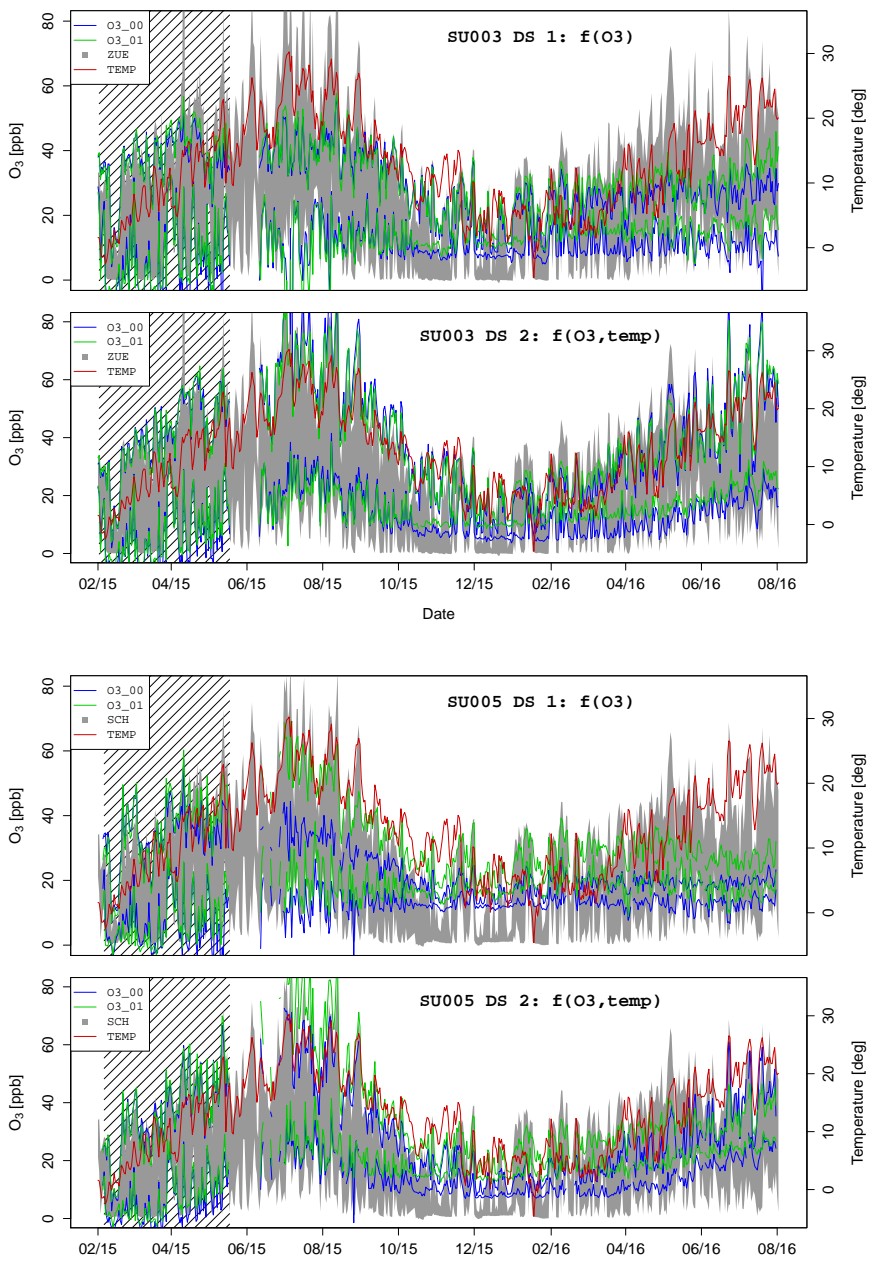

**Figure 5.** Daily range of 30 minutes mean $O_3$ concentrations indicated by the two sensors O3_00 and O3_01 integrated in the SUs 003 and 005. It is depicted by two lines, respectively, representing the daily maximum and minimum (DS 1: basic linear model; DS 2: advanced model based on Eq. 5 including temperature as a parameter; see table 4 for details). The gray band depicts the same value for the AQM site where the sensor unit was located for calibration. This AQM site is also comparable with the location where the SU was operated within the sensor network with respect to the pollutant situation. The time period when the sensors were calibrated at the AQM site is depicted by a shaded area (calibration: "PAR"). Temperature is the daily mean value at AQM site ZUE.





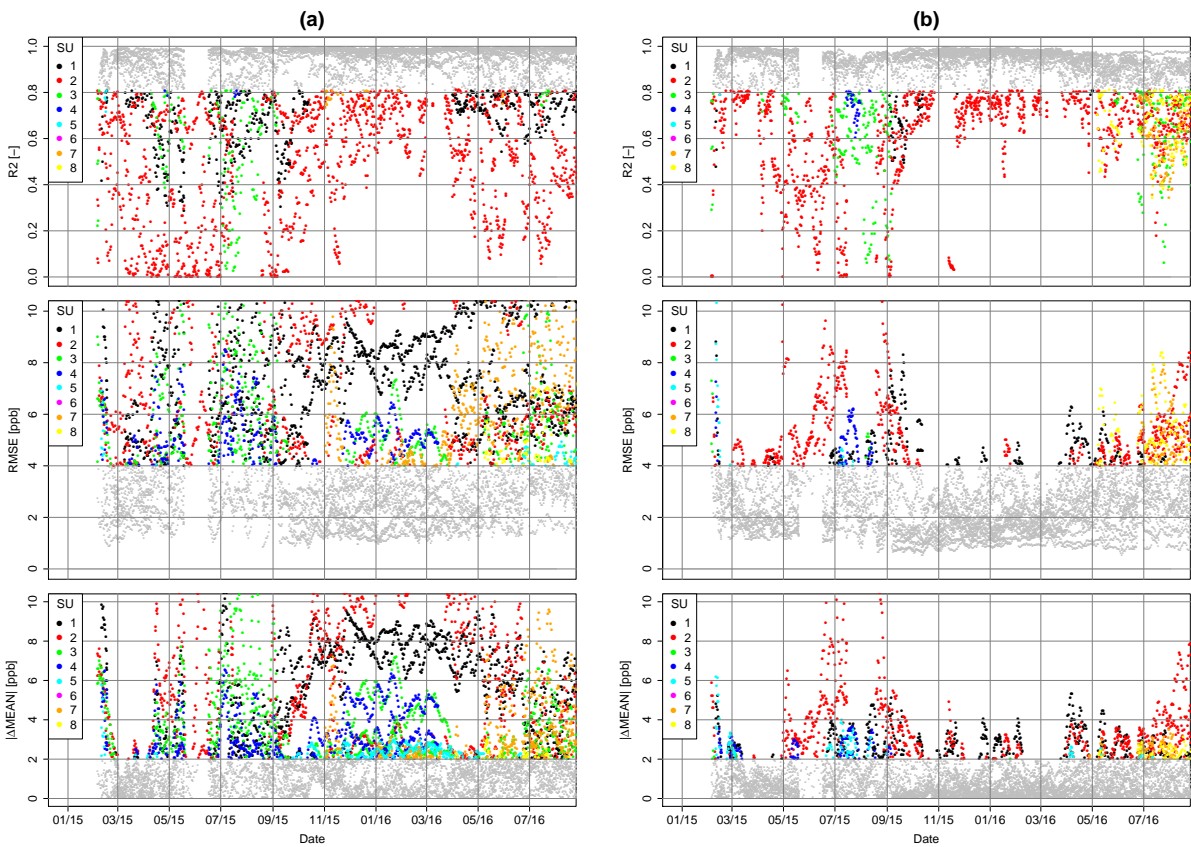

**Figure 6.** Comparison of measurements from the redundant $NO_2$ sensors integrated in the SUs by means of the 7 days rolling RMSE, R2 and absolute average difference based on 30 minutes $NO_2$ concentrations. There are $n(n-1)/2$ values for $n$ integrated sensors per day. Only time periods are considered when the SUs were calibrated or operated in the sensor net. The values in (a) are derived from data set 1 which is based on a linear fit only. The values in (b) are derived from data set 3 (calibration "PAR/REM"; based on Eq. 6). Corresponding results for the $O_3$ sensors are included in the supplementary materials.





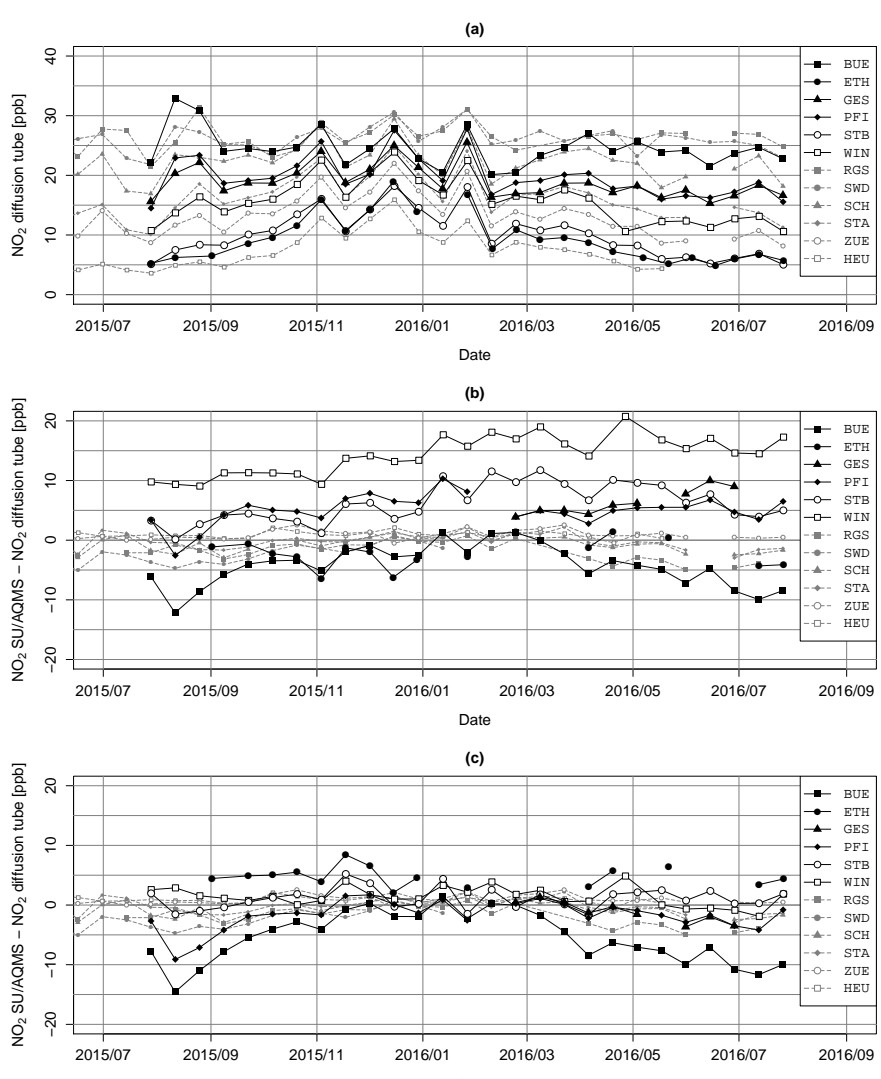

**Figure 7.** (a) Two-week NO$_2$ diffusion tube measurements. (b) and (c) Differences of two-week NO$_2$ mean values provided by SUs (DS 2 in (b), DS 3 in (c)) and by diffusion tubes.

.




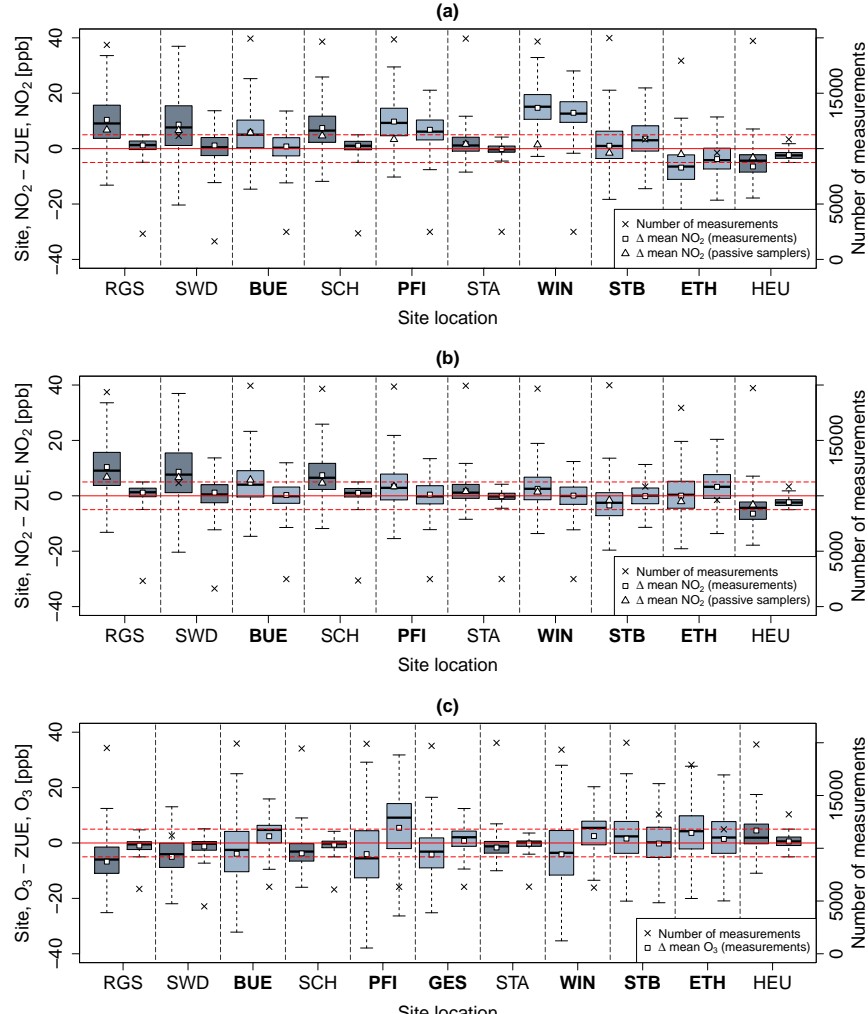

**Figure 8.** Analysis of the differences of $NO_2$ and $O_3$ measurements from SU/AQM sites and measurements from the AQM site ZUE. The comparison is twofold: for each site, the boxes on the left include all measurements while the boxes on the right only include the measurements when concentration ranges are below 5 ppb for the AQM site groups *background* and *city*. Measurements of data set 2 are used in (a). Measurements of data set 3 are used in (b) and (c). The site names are bold for SU locations. Note that the sensor network locations STB and ETH refer to the group *background* and the locations BUE, PFI, GES and WIN refer to the group *city*

.





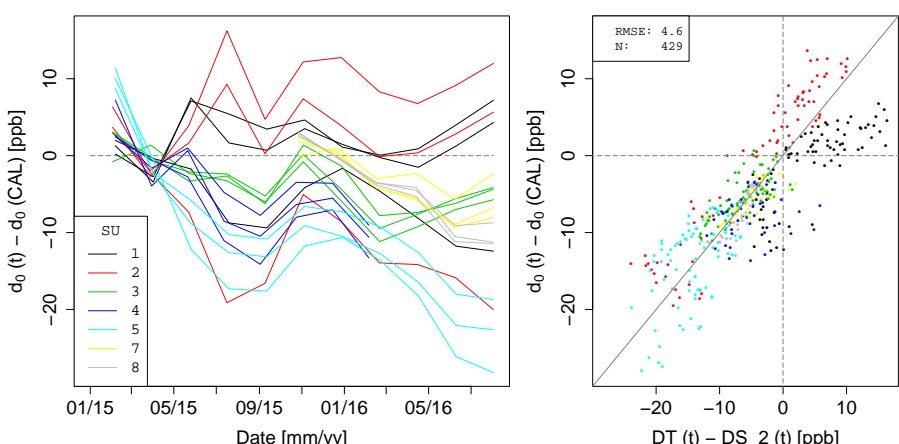

**Figure 9.** (a) Time series of the piece-wise linear function $d_0$ of Eq. 6 (data set 3: calibration "PAR/REM") with the average $d_0$ value during the initial calibration phase subtracted. (b) Comparison of the values from (a) with the difference between two-week $NO_2$ concentrations from diffusion tubes (DT) and measurements from the $NO_2$ sensors (data set 2: Eq. 6 with $d_0 = const$, calibration "PAR"). The depicted solid line shows the 1:1 ratio.

.





**Figure 10.** Comparison of the measurements from sensors integrated in SU004 and SU006 (data set 2) to measurements from the AQM site SCH during the 2$^{nd}$ calibration in the period February to March 2016. Top row: $O_3$ sensor no. 0 of SU004 and SU006, respectively. Middle row: $NO_2$ B42F sensors no. 0 and 1 of SU004. Bottom row: $NO_2$ B4 sensors no. 0 and 1 of SU006. The selected sensors are representative.




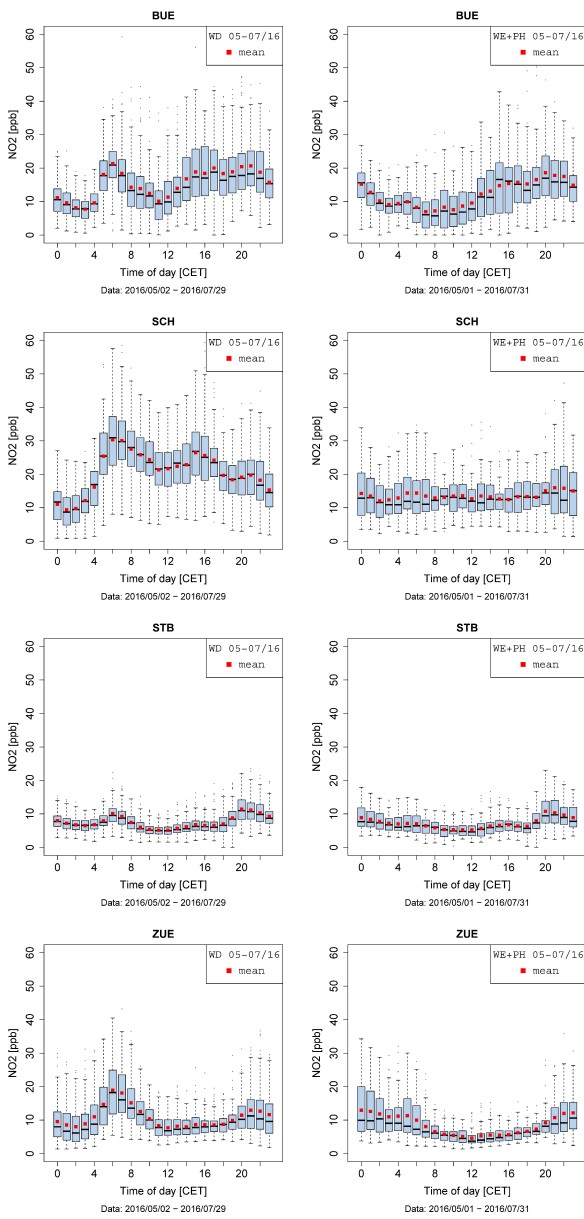

**Figure 11.** Comparison of the diurnal variation of NO$_2$ concentration at two AQM sites (SCH and ZUE) and at two sensor network locations (BUE and STB, data of DS 3). The left-hand figures depict data from working days (WD), the right-hand figures data from weekends and public holidays (WE+PH), respectively. Sites RGS and BUE are roadside locations, sites STB and ZUE are background locations.