# Peer review of "Design of an ozone and nitrogen dioxide sensor unit and its long-term operation within a sensor network in the city of Zurich"

_Atmospheric Measurement Techniques, 2017_

## Referee Comment (RC1) · Anonymous Referee #2 · 4 Apr 2017

General Comments This study presents long-term measurements with low-cost ozone and NO2 sensors, and reports on their accuracy, sensitivity as well as providing a method for their calibration and data control for long-term measurements. I found it an interesting and well thought out study, with some interesting findings related to their sensor performance when deployed for ambient monitoring. There are not many studies reporting on the long-term deployment of low cost sensors and so their finding would be of broad interest as low cost sensors is a hot topic. While the paper was mostly well written, I did find some sections a bit hard to read, and I think that this could be improved by being more explicit at the end of sections on what the actual findings/conclusions were. For example, by the end of Section 5.2, it wasn't clear to

me whether the sensors were in agreement? Or Section 4.2, which was the better model to use? I think by adding a sentence or two at the end would help the reader follow the progression of findings through the paper. I would add though that I did find the overall findings and conclusion well summarized in the final section. While there are not many, there have been some reported studies on ambient measurements using 'low-cost' sensors, and I would have liked to see some discussion comparing results to the literature. For example, Lewis et al., (2016) found seemingly contradictory results, as they found the ozone sensor performed better than the NO2 sensor for ambient measurements and I would interested to on the authors thoughts on why.

Overall, I would recommend this manuscript for publication after consideration to the following comments.

Specific comments

Abstract, line 9: Perhaps you could indicate the accuracy of the diffusion tubes?

Abstract: It would be worth also mentioning how the ozone sensors performed, a part from their accuracy?

Section 2.2.1: It was not clear to me where the sites used where located? I suggest amending Table 1 to include how far the AQM/reference sites were from the sensor units, this is important information when understanding the calibrations and corrections applied later.

Section 3.1: Why were only 2 ozone sensors used and not three like for NO2? How can you be sure which is ozone sensor was correct?

Section 3.3, line 11? What does  $\Delta t0$  represent and how exactly did you measure it in the field?

Section 3.3, line 18-19: I think you should expand this discussion, is just RH and NO2 that the sensor is responding to?

Section 4.1: I found the description of the different data sets a bit confusing. Was the 10% of the data selected from the whole time series for the calibration(whole year)? The reported concentrations for NO2 and O3, were these for the 10% of the selected data for the calibration? I suggest that this paragraph is re-worded to clarify what data sets were used for calibration of each model.

Section 4.2, line 26: Which variables were used in the model?

Page 9, line 20: Please re-word, I wasn't sure what became evident of the ozone sensors.

Section 5.1, line 6: Should you remove the negative values? Wont this give a positive bias to your averaging?

Section 5.3: In Fig 7, there appears to be a seasonal trend, with greater discrepancy between the sensors and the diffusion tubes during the summer compared to winter? Perhaps the authors could consider why this may have occurred?

Section 5.5: Why did the ozone sensors have such poor agreement at the end when the NO2 sensors did not have this issue? You mentioned earlier the issue of them being clogged by airborne particles, was this a contributing factor? How come the agreement was poor when the ozone sensors were still reasonably correlated with the reference instrument? In addition, it is also not clear here how the measurements at AQM differ to the reference instruments?

Figure 2: I found this figure hard to understand; it wasn't clear to me how it depicts 3 tests, perhaps because the y-axis has many parameters. I suggest simplifying by only including the most relevant data, or splitting into multiple plots.

Figure 3: What do the two columns of numbers on the left of the plots represent? I would also indicate what RMSE represents in the caption.

Technical Comments: Page 6 line 21: Should it read 'may both experience interference with temperature and humidity'?

Page 7 line 8: should it read: 'calibration of the sensors for all the SUs'

Page 8, line 14: Obviously is mis spelt.

Page 9, line 18: Progressively rather than progressionally?

Page 10, line 18-19: Please indicate the figure number, I'm guessing 6?

Page 11: line 26-7: please indicate the figure numbers.

References

Lewis, A.C., Lee, J.D., Edwards, P.M., Shaw, M.D., Evans, M.J., Moller, S.J., Smith, K.R., Buckley, J.W., Ellis, M., Gillot, S.R., White, A., 2016. Evaluating the performance of low cost chemical sensors for air pollution research. Faraday Discussions 189, 85-103.

---

## Referee Comment (RC2) · Anonymous Referee #3 · 27 Apr 2017

This is a very thorough and very timely publication that examines the performance of NO2 and ozone electrochemical sensors when used as part of an operation air quality network in Zurich. There is very high demand for work of this kind, and the literature contains rather few balanced studies that take a long-term approach to evaluating performance. There is some basic lab testing reported, comparisons of various correction models, and some interesting approaches to data analysis, including comparisons of a sensor network during periods of notionally homogeneous atmospheric composition.

The manuscript is rather dense and technical to read so would benefit from some careful editing to improve readability. There are a few areas where some additional clarifications are needed, but in principle the paper should be published in close to its

current form.

The sensor units developed by the EMPA team are described in some detail, each containing a number of duplicate and triplicate sensors. The text isn't very clear however about how data from the redundant sensors is used in the paper's subsequent analysis. Is the information from all three NO2 sensors included in the statistics on for example RMSE (mean, median?), or is one single sensor chosen chosen, with the other two NO2 sensors truly redundant spares in case of failure? The later part of the paper shows how the identical sensors in each box compare to one another and this is very interesting, but there needs to be better clarification of how each contributes to the datasets that are the main conclusions of the study. Page 10 refers to the mean of sensors in each box, but its not clear if this approach is used through the paper, or just in this part.

The paper includes some lab testing of sensor interferences, and this shows some similar results to other studies. Are the various mixtures of co-pollutants that are tested presented to the sensors in 'real' air (e.g. zero air, or synthetic air) or blended in pure N2?

Figure 2 is really far too small to be read clearly for so many different chemicals, so a better way of showing this data is needed.

There is a lot of information in Figure, and the text refers to this figure as showing for example the impacts and temperature RH correction compared between models. It is quite hard for the reader to find this in a many different models, so could the figure or text more directly identify those models that show these differences?

The comparisons between sensors in Figure 6 are interesting (associated text is on page 10). There is reference to drift in RMSE and R2 over time, but its quite hard to see this in the data. I presume this is inferred because towards the end of the period more sensors have RMSE value above 4 ppb, than at the start? Is there perhaps another way this could be shown graphically since this is an important point.

The comparison of sensors in homogeneous air is an interesting approach. Can the figure and text be made a little clearer about which bits of data in Figure 8 are from Sensor Units, and which are data being reported from standard reference instruments. I have assumed that the sites in normal font, eg SWD, SCH, STA are only showing reference instrument data, for those selected periods, whilst WIN, STV, ETH for example are only showing sensor data?

Figure 11 seemed to be a little surplus to requirements in the paper. It is good that the Sensor Units captured a plausible diurnal trend, but since the two sensor locations and two AQ reference stations are not co-located, there isn't much to infer from comparing the two types of data.

Minor editorial changes

Page 2 'metal oxide'

Page 2 Line 27, '. . ..were operated at these locations until August 2. . ..

Page 5. Line 11. Presumably this should read something like: '. . .leading to the omission of a few measurements where there were small variations in measurement frequency'?

Page 6 Line 4. It is temperature and humidity that interfere with the sensors, not the other way around.

Page 7. Line 7. This isn't clear, but I have assumed this to mean the sensors were operated some way away from the reference site. The explanation of the PAR/REM approach to calibration needs a slightly expanded and better description here. It becomes clearer the more you read on in latter pages.

P8 line 14 – obviously

Figure 6 (and elsewhere). Can the captioning use the same abbreviations as the text, eg DS1, DS3. There is some interchange between dataset 2 - DS 2 etc.

---

## Author Comment (AC1) · 27 Jun 2017

The comment was uploaded in the form of a supplement:
https://www.atmos-meas-tech-discuss.net/amt-2017-22/amt-2017-22-AC1-supplement.zip

---

## Author Response (AR1)

**Reply to the comments of reviewer #2 on the manuscript „Design of an ozone and nitrogen dioxide sensor unit and its long-term operation within a sensor network in the city of Zurich"**

Michael Müller[1], Jonas Meyer[2] and Christoph Hüglin[1]

[1]Empa, Swiss Federal Institute of Materials Science and Technology, Dübendorf, Switzerland.

[2]Decentlab Gmbh, Dübendorf, Switzerland.

We thank the reviewer for his effort in evaluating this manuscript and for his valuable suggestions for improvements. All points made by the reviewer are addressed on the following pages.

**Section 5.2. Sensors in agreement?**

**Response:** We state in this section that the sensors, in general, are coherent and that time-dependent parameters in the sensor model reduce RMSE and absolute difference between the sensors. This is the main statement. Corresponding numeric values are shown in Figure 6.

**Modification:** We made some changes in this section based on a comment of reviewer #3.

**Section 4.2. Better model to use?**

**Modification:** We added a sentence at the end of this section referring to table 4 where under DS3 the models are summarized that we think performed best.

**Reviewer #2 asks to comment why $O_3$ sensors perform better than $NO_2$ sensors in the study of Lewis et al. (2016)**

**Response:** Lewis et al. (2016) tested in their study five different types of electrochemical sensors from Alphasense ($CO_2$, $O_3$, $NO$, $NO_2$ (B4), $SO_2$). In our study electrochemical Alphasense $NO_2$ (B4, B42F) and Aeroqual metal oxide $O_3$ sensors were tested. Sensors for particular species produced by certain manufacturers differ by working principles, design and electronics and have to be analysed specifically. In our opinion, the studies are not contradictory but complement each other.

**Modification:** We included the working principles of the $O_3$ and $NO_2$ sensors in the beginning of section 3.1.

**Abstract, line 9: Perhaps you could indicate the accuracy of the diffusion tubes?**

**Modification:** We included the accuracy of the diffusion tubes in the abstract.

**Abstract: It would be worth also mentioning how the ozone sensors performed, a part from their accuracy?**

**Modification:** We added a sentence reporting that the sensitivity of the $O_3$ sensors decreased over time.

**Section 2.2.1: It was not clear to me where the sites used where located? I suggest amending Table 1 to include how far the AQM/reference sites were from the sensor units, this is important information when understanding the calibrations and corrections applied later.**

**Modification:** We included in table 1 the distance of the locations of the sensor network to the nearest AQM site as well as the distance to AQM site ZUE. In addition, we refer in the legend of table 1 to the map included in the supplementary materials showing the AQM sites and the network location sites.

**Section 3.1: Why were only 2 ozone sensors used and not three like for NO2? How can you be sure which ozone sensor was correct?**

**Response:** Actually, in first field test measurements the $O_3$ sensors showed a high performance and proofed to work reliably. It only showed during longer-term deployment that the $O_3$ sensors suffered from a changing response behaviour. Taking this prior experience and the costs for redundant sensors into account, we decided to work with a two ozone sensor configuration. Ozone sensors of two sensor units exhibited largely differing ozone values during operation. In these cases we focused on the sensor providing the more plausible measurements with respect to measurements from AQM sites.

The reviewer is right that with a two sensor configuration the malfunctioning device cannot be identified if there are differences between the two sensors and both sensors report plausible values. However, correspondence of measurements from two sensors does not necessarily mean that the sensors work properly neither. They just might have been impacted by identical environment factors leading to degradation of the sensors. Therefore, we pointed out in the manuscript that a sensor performance analysis strategy is necessary for the operation of a sensor network.

**Modification:** No changes.

**Section 3.3, line 11? What does _t0 represent and how exactly did you measure it in the field?**

**Response:** The parameter $\Delta t0$ represents the time when the impact of a particular change in relative humidity on the sensor signal decayed to 0.37 of its initial value (see Eq. 2). We did not measure the parameter but computed several models with different values for $\Delta t0$. We set $\Delta t0$ to the value of the best performing model.

**Modification:** We added a sentence in section 3.3 explaining the method of finding the optimal value for t0.

**Section 3.3, line 18-19: I think you should expand this discussion, is just RH and $NO_2$ that the sensor is responding to?**

**Response:** As stated we did not found any cross-sensitivities of the $NO_2$ sensors to other gases in our experiments. The (slight) impact of temperature on the signal is discussed in section 4.3. Therefore, we do not see how we could substantially expand the discussion.

**Modification:** No changes.

**Section 4.1: I found the description of the different data sets a bit confusing. Was the 10% of the data selected from the whole time series for the calibration(whole year)? The reported concentrations for $NO_2$ and $O_3$, were these for the 10% of the selected data for the calibration? I suggest that this paragraph is re-worded to clarify what data sets were used for calibration of each model.**

**Modification:** We reworded section 4.1. in order to improve clarity.

**Section 4.2, line 26: Which variables were used in the model?**

**Response:** That is already explained in section 4.2., page 8, lines 1 to 9.

**Modification:** We added a sentence after discussion of the sensor model performance in section 4.3 that in table 4 the models are specified which we consider to provide the best data.

**Page 9, line 20: Please re-word, I wasn't sure what became evident of the ozone sensors.**

**Modification:** We reworded the sentence.

**Section 5.1, line 6: Should you remove the negative values? Won't this give a positive bias to your averaging?**

**Response:** The "sensor unit $NO_2/O_3$ measurement" $S_u(t_i)$ for a one minute interval is computed based on the particular $NO_2/O_3$ sensor measurements $S_1(t_i)$, $S_2(t_i)$, $S_3(t_i)$. Although negative concentrations do not exist negative sensor values $S_i(t_i)$ may result for an individual sensor if the applied statistical sensor model does not entirely capture the physical sensor behaviour.

For the computation of 30 minutes mean "sensor unit measurements" $S_u(30\ min)$ negative one minute "sensor unit measurements" $S_u(t_i)$ were treated as zero concentrations. As the reviewer points out the average value obtained with this procedure is equal (in case of non-negative averages only) or larger than that obtained without equating to zero negative values. We do not expect that the error term associated with negative values is cancelled out for 30 minutes averages. So the procedure of setting negative values to zero most likely leads to more accurate values.

During manuscript preparation it has been forgotten that we set negative $O_3$ sensor values to zero for the computation of the RMSE used for the comparison of different $O_3$ sensor models. We treated $NO_2$ and $O_3$ differently by two reasons: First, we thought at the beginning of the study the $O_3$ sensors might be usable for correcting the $O_3$ cross-sensitivity of the $NO_2$ B4 sensors. The use of negative concentrations would not have been appropriate in this context. Second, we used only $O_3$ values larger than 2 ppb for the estimation of the sensor model parameters. However, the statements in the manuscript do not change.

**Modification**: We reworded section 5.1 accordingly. We specify more accurately the computation of the RMSE in section 4.3 and in Figure 4.

**Section 5.3: In Fig 7, there appears to be a seasonal trend, with greater discrepancy between the sensors and the diffusion tubes during the summer compared to winter? Perhaps the authors could consider why this may have occurred?**

**Response:** For the sensor unit located at BUE large (>5 ppb) and long-term discrepancies with respect to $NO_2$ diffusion tubes were encountered in the summer seasons. Location BUE is next to a busy road and close to Lake Zurich (< 100 m).
Diffusion tubes may exhibit dependencies on external factors such as temperature, humidity or wind speed. The used diffusion tubes as well as the mountings were identical at all locations. The comparison between the diffusion tubes and the measurements from the reference instruments showed no large differences for this time period. An issue with the diffusion tubes is not obvious.
We have no meteorological measurements for site BUE and the other locations of the sensor network except for the temperature and humidity sensors inside the sensor units. Thus, knowledge of the differences in environment conditions is limited.

In summary, we think that the data basis is not sufficient for substantially resolving the cause for the differences between the sensor and diffusion tube measurements.

**Modification:** No changes.

**Section 5.5: Why did the ozone sensors have such poor agreement at the end when the NO₂ sensors did not have this issue? You mentioned earlier the issue of them being clogged by airborne particles, was this a contributing factor? How come the agreement was poor when the ozone sensors were still reasonably correlated with the reference instrument? In addition, it is also not clear here how the measurements at AQM differ to the reference instruments?**

**Response:** The Alphasense NO$_2$ sensor is an electrochemical sensor whereas the Aeroqual O$_3$ sensor is a metal oxide sensor. They exhibited almost identical environment conditions during operation but cannot directly be compared as they are based on different working principles.

Figure 10 shows that the NO$_2$ sensors measurements (DS 2) of two SU are reasonably in agreement with measurements from reference instruments after more than one year of operation while the O$_3$ sensors are not. This study focuses on data analysis. We have no detailed information about the design of the sensors. Therefore, we are limited in the interpretation of our findings related to technical aspects of the sensors. We reported our observations to the manufacturer and received several suggestions. We refrain from further interpretations as they would remain speculations.

**Modification:** We included the working principles of the O$_3$ and NO$_2$ sensors in the beginning of section 3.1.

**Figure 2: I found this figure hard to understand; it wasn't clear to me how it depicts 3 tests, perhaps because the y-axis has many parameters. I suggest simplifying by only including the most relevant data, or splitting into multiple plots.**

**Modification:** We split Figure 2 (a) into four subplots in order to improve clarity.

**Figure 3: What do the two columns of numbers on the left of the plots represent? I would also indicate what RMSE represents in the caption.**

**Response:** The meaning of these numbers is explained in the last line of the figure caption.

**Modification:** We added the meaning of RMSE in the figure caption.

**Technical Comments:**
Page 6 line 21: Should it read 'may both experience interference with temperature and humidity'?
➔ Corrected.
Page 7 line 8: should it read: 'calibration of the sensors for all the SUs'
➔ We slightly changed the sentence in order to improve clarity.
Page 8, line 14: Obviously is mis spelt.
➔ Corrected.
Page 9, line 18: Progressively rather than progressionally?
➔ Corrected.
Page 10, line 18-19: Please indicate the figure number, I'm guessing 6?
➔ Figure number 6 is already written on page 10, line 18. We changed a "the" to "this" in order to make clear we refer to the same figure again.
Page 11: line 26-7: please indicate the figure numbers.
➔ Figure number 9 is already written on page 11, line 26. We changed a "the" to "this" in order to make clear we refer to the same figure again.

**Reply to the comments of reviewer #3 on the manuscript „Design of an ozone and nitrogen dioxide sensor unit and its long-term operation within a sensor network in the city of Zurich"**

Michael Müller[1], Jonas Meyer[2] and Christoph Hüglin[1]

[1]Empa, Swiss Federal Institute of Materials Science and Technology, Dübendorf, Switzerland.

[2]Decentlab Gmbh, Dübendorf, Switzerland.

We thank the reviewer for his effort in evaluating this manuscript and for his valuable suggestions for improvements. All points made by the reviewer are addressed on the following pages.

**The text isn't very clear however about how data from the redundant sensors is used in the paper's subsequent analysis. Is the information from all three NO2 sensors included in the statistics on for example RMSE (mean, median?), or is one single sensor chosen, with the other two NO2 sensors truly redundant spares in case of failure? The later part of the paper shows how the identical sensors in each box compare to one another and this is very interesting, but there needs to be better clarification of how each contributes to the datasets that are the main conclusions of the study. Page 10 refers to the mean of sensors in each box, but it is not clear if this approach is used through the paper, or just in this part.**

> **Response:** We agree with the reviewer that the reader should easily understand which part of the manuscript refers to measurements from individual sensors and in which parts the mean of all the sensors in a sensor unit has been used. This was not always clear.

> **Modification:** We therefore added a paragraph at the end of section 3.1 that explains in which section individual sensors are discussed and in which sections the discussion refers to the mean. This information is repeated in section 5.1 as in section 5 discussions refer to both individual sensor measurements and the mean of the individual sensor measurements.

**The paper includes some lab testing of sensor interferences, and this shows some similar results to other studies. Are the various mixtures of co-pollutants that are tested presented to the sensors in 'real' air (e.g. zero air, or synthetic air) or blended in pure N2?**

> **Response:** NO, NO2, CO, CO2 were added in specific concentrations to conditioned zero air.

> **Modification:** We added this information in the text.

**Figure 2 is really far too small to be read clearly for so many different chemicals, so a better way of showing this data is needed.**

> **Modification:** We optimized Figure 2 in order to present our findings in a clearer way.

**There is a lot of information in Figure, and the text refers to this figure as showing for example the impacts and temperature RH correction compared between models. It is quite hard for the reader to find this in a many different models, so could the figure or text more directly identify those models that show these differences?**

**Response:** We assume the reviewer is referring to Figure 3. The figures are linked with equations 2 and 3 to 6 in terms of the variable naming. The discussion about the model performance is also based on these equations. We are aware that there is a lot of information in Figure 3, nevertheless Figure 3 is complete and provides all the information needed for the assessment of the dependence of the model performance on the selection of the different terms in the sensor models.

**Modification:** We suggest keeping Figure 3 as it is.

**The comparisons between sensors in Figure 6 are interesting (associated text is on page 10). There is reference to drift in RMSE and R2 over time, but its quite hard to see this in the data. I presume this is inferred because towards the end of the period more sensors have RMSE value above 4 ppb, than at the start? Is there perhaps another way this could be shown graphically since this is an important point.**

**Response:** The large differences between the mean differences and RMSE values between Figure 6 (a) and Figure 6 (b) can clearly be identified. The term "drift" is not entirely adequate. We agree with the reviewer.

**Modification:** We changed a sentence in section 5.2 in order to focus on the main message of Figure 6. In addition, we limited the time axis of the figure to August 2, 2016 as this is the end of the data period used in the sensor calibration.

**The comparison of sensors in homogeneous air is an interesting approach. Can the figure and text be made a little clearer about which bits of data in Figure 8 are from Sensor Units, and which are data being reported from standard reference instruments. I have assumed that the sites in normal font, eg SWD, SCH, STA are only showing reference instrument data, for those selected periods, whilst WIN, STV, ETH for example are only showing sensor data?**

**Response:** It is correct the site names in normal font refer to measurements from reference instruments and the site names in bold refer to sensor measurements.

**Modification:** We changed the caption of Figure 8 in order to improve the explanation of the figure content. We think it is much clearer now.

**Figure 11 seemed to be a little surplus to requirements in the paper. It is good that the Sensor Units captured a plausible diurnal trend, but since the two sensor locations and two AQ reference stations are not co-located, there isn't much to infer from comparing the two types of data.**

**Response:** Traffic causes most $NO_2$ emissions in Zurich. Diurnal variation of traffic volume in the city centre of Zurich is not homogeneous but comparable. Therefore, we think that locations impacted by similar traffic volume can qualitatively be compared. This plot is valuable in order to give the reader an impression of the obtained data quality which is encouraging (as indicated in this plot) but not yet what is required for long-term use (comparison with diffusion tubes).

**Modifications:** We therefore suggest keeping this Figure in the manuscript and not to remove it from the paper.

**Minor editorial changes**
Page 2 'metal oxide'

➔ Corrected.

Page 2 Line 27, ': : :.were operated at these locations until August 2: : :.

➔ Corrected.

Page 5. Line 11. Presumably this should read something like: ': : :leading to the omission of a few measurements where there were small variations in measurement frequency'?

➔ Changed.

Page 6 Line 4. It is temperature and humidity that interfere with the sensors, not the other way around.

➔ Corrected.

Page 7. Line 7. This isn't clear, but I have assumed this to mean the sensors were operated some way away from the reference site. The explanation of the PAR/REM approach to calibration needs a slightly expanded and better description here. It becomes clearer the more you read on in latter pages.

➔ We reworded section 4.1 in order to better explain the calibration concept.

P8 line 14 – obviously

➔ Corrected.

Figure 6 (and elsewhere). Can the captioning use the same abbreviations as the text, eg DS1, DS3. There is some interchange between dataset 2 - DS 2 etc.

➔ We made several changes in the text to be more coherent concerning naming of the data sets.